# Impact of the COVID-19 Pandemic on Maternal Anxiety in Brazil

**DOI:** 10.3390/jcm10040620

**Published:** 2021-02-06

**Authors:** Roseli Nomura, Isabela Tavares, Ana Carla Ubinha, Maria Laura Costa, Maria Lucia Opperman, Marianna Brock, Alberto Trapani, Lia Damasio, Nadia Reis, Vera Borges, Alberto Zaconeta, Ana Cristina Araujo, Rodrigo Ruano

**Affiliations:** 1Department of Obstetrics, Escola Paulista de Medicina, Federal University of São Paulo, Rua Napoleão de Barros, 875, Sao Paulo, SP CEP 04024-002, Brazil; isatavares.epm@gmail.com (I.T.); ana.ubinha@gmail.com (A.C.U.); 2Department of Obstetrics and Gynecology, University of Campinas, Campinas, SP CEP 13083-881, Brazil; mlaura@unicamp.br; 3Gynecology and Obstetrics Unit, Hospital de Clínicas de Porto Alegre, Department of Gynecology and Obstetrics, Faculdade de Medicina, Federal University of Rio Grande do Sul, Porto Alegre, RS CEP 90035-903, Brazil; maluopp@gmail.com; 4Department Obstetrics and Gynecology, University of Amazonas State, Manaus, AM CEP 69065-001, Brazil; mariannabrock@hotmail.com; 5Women’s Health Care Unit, Polydoro Ernani de São Thiago University Hospital, Federal University of Santa Catarina, Florianópolis, SC CEP 88036-800, Brazil; ginecoalberto@yahoo.com.br; 6Department Obstetrics and Gynecology, Federal University of Piaui, Teresina, PI CEP 64049-550, Brazil; liacruzcosta@yahoo.com.br; 7Gynecology and Obstetrics Unit, University Hospital Maria Aparecida Pedrossian of Medical School, Federal University of Mato Grosso do Sul, Campo Grande, MS CEP 79080-190, Brazil; nsreis@hotmail.com; 8Department Obstetrics and Gynecology, Botucatu Medical School- Unesp, Botucatu, SP CEP 18618-687, Brazil; vborges0765@gmail.com; 9Maternal and Child Unit, University Hospital of Brasília, Faculty of Medicine, University of Brasília, Brasília, DF CEP 70910-900, Brazil; azaconeta@gmail.com; 10Department of Gynecology and Obstetrics, Maternidade Januário Cicco, Federal University of Rio Grande do Norte, Natal, RN CEP 59078-900, Brazil; anacrys.araujo@gmail.com; 11Maternal-Fetal Medicine Division, Department of Obstetrics and Gynecology, Mayo Clinic College of Medicine, Rochester, MN 55905, USA; ruano.rodrigo@mayo.edu

**Keywords:** pregnancy, maternal anxiety, childbirth, postpartum, questionnaires, breastfeeding, coronavirus disease 2019, pandemic

## Abstract

Background: The coronavirus disease 2019 (COVID-19) pandemic may have an effect on maternal anxiety and influence postpartum experience. Objective: To study the prevalence of maternal anxiety in late pregnancy in the context of the COVID-19 outbreak in Brazil and to analyze its association with maternal knowledge and concerns about the pandemic. Methods: This was a national multicenter cross-sectional study performed in 10 different public university hospitals, between 1 June and 31 August 2020, in Brazil. The inclusion criteria were: maternal age more than 18 years; gestational age more than 36 weeks at childbirth; single alive newborn without malformations; and absence of mental disorders. We applied a structured questionnaire to explore the knowledge and concerns about COVID-19. Maternal anxiety was assessed using the Beck Anxiety Inventory (BAI). Results: Of the 1662 women interviewed, the BAI score in late pregnancy indicated that 13.9% presented moderate and 9.6% severe maternal anxiety. Moderate or severe maternal anxiety was independently associated with the fear of being unaccompanied at childbirth (aOR1.12, 95% CI 1.10–1.35), and independent protective factors were confidence in knowing how to protect oneself from COVID-19 (aOR0.89, 95% CI 0.82–0.97) and how to safely breastfeed (aOR0.89, 95% CI 0.83–0.95). Conclusion: The COVID-19 pandemic has a significant impact on maternal anxiety.

## 1. Introduction

The coronavirus disease 2019 (COVID-19) outbreak represents a major source of stress for pregnant women [1]. This pandemic is a challenge worldwide and interventions to minimize the impact on people’s quality of life are imperative. Pregnant women have had to face various factors during the pandemic, including quarantine policies, social isolation, unemployment, fear of falling ill and having the disease affect loved ones. All of these factors may negatively impact women’s mental well-being.

On 31 December 2019, the World Health Organization (WHO) received a notification of cases of atypical pneumonia in the city of Wuhan, China [2]. The first case of this disease was followed by an epidemic that rapidly took on global proportions. COVID-19, caused by severe acute respiratory syndrome coronavirus 2 (SARS-CoV-2), was declared a pandemic by WHO on 11 March 2020 [3]. In Brazil, the first case of COVID-19 was detected on 26 February 2020, in São Paulo (Southeast region). With the increase in the number of cases and deaths in the country, in April 2020, the health system of Manaus (northern region) collapsed. In May, some of the cities in the Northeast were placed under lockdown. On 22 May, due to the rapidly growing number of infections in Brazil, the WHO declared South America the new epicenter of the pandemic. The highest recorded daily number of new cases (69,074 cases) and deaths (1595 deaths) occurred on 29 July [4]. High prevalence of depression (61%) and anxiety (44%) was found in a national online survey applied to the Brazilian population during the SARS-CoV-2 outbreak [5]. Among pregnant women evaluated in primary care facilities in Sao Paulo, reported anxiety prevalence was 20.4% before the COVID-19 pandemic [6].

This outbreak is leading to additional worldwide health problems, such as stress, anxiety, depressive symptoms, insomnia, denial, anger and fear [7,8,9,10,11]. The COVID-19 pandemic has also increased anxiety among the pregnant population [12]. Corbett et al. [13] found that pregnant women were the most concerned about their elderly relatives (83%), followed by concerns related to their children (67%) and their unborn child (63%). Women were less concerned about their own health; nevertheless, many of them were significantly anxious. In Wuhan, China, the COVID-19 outbreak increased pregnant women’s anxiety and affected their decision-making on prenatal care schedules or timing of childbirth, mode of delivery and infant feeding [14]. Wu et al. [15] reported depressive symptoms in 29.6% of pregnant women after the epidemic was declared. In Belgium, an online survey during the lockdown period revealed that 14% of pregnant and breastfeeding women met the criteria for high anxiety [16]. Interventions targeting maternal stress and isolation, such as effective communication and psychological support, should be offered to decrease such mental health effects.

The effects of the pandemic in low- and middle-income settings are underreported. Mental health in pregnant women during the period of greatest mortality in Brazil due to COVID-19 has not been assessed. Therefore, the aim of the present study was to evaluate the prevalence of maternal anxiety in late pregnancy in the context of the COVID-19 outbreak in Brazil and to investigate the association with sociodemographic characteristics and maternal knowledge and concerns about the pandemic.

## 2. Material and Methods

This multicenter cross-sectional study was conducted in 10 cities in all five Brazilian regions. This study was approved by each local Research Ethics Committee and by the Brazilian National Ethics Committee—Conep (No. of approval: CAAE-31190120.6.1001.5505). A written informed consent statement was signed by every woman included in the study.

### 2.1. Women’s Enrolment

Women who delivered in 10 public university hospitals were recruited from 1 June 2020 to 31 August 2020. Enrolment took place for a period of 60 consecutive days at each center. All hospitals were linked to a federal or state public university located in 10 cities in all five geographical regions of Brazil: North (Manaus), Northeast (Natal and Teresina), Southeast (São Paulo, Campinas and Botucatu), South (Florianópolis and Porto Alegre) and Central West (Campo Grande and Brasília) (Figure 1).

Every university center involved in this study had a local coordinator, who was a professor at the Department of Obstetrics and Gynecology. Medical residents involved in data collection were trained to conduct the interviews and to guide women in completing the forms. Participants were counseled on their concerns about protective measures to minimize COVID-19 exposure; and cases that required psychological care were referred for evaluation and follow-up with specialists available at each facility.

Inclusion criteria were the following: maternal age more than 18 years; gestational age at delivery more than 36 weeks; single alive newborn without malformations; no clinical suspicion or current diagnosis of COVID-19; absence of diagnosed psychiatric or mental disorders (such as depression, bipolar disorder, schizophrenia, etc.); and in good health, in the absence of severe obstetrical/clinical complication during childbirth or intensive care unit (ICU) admission. Those who agreed to participate were enrolled. Trained medical residents interviewed postpartum women up to the third day after childbirth before hospital discharge.

### 2.2. Questionnaires’ Application

Participants were asked to complete a sociodemographic questionnaire. The questions addressed maternal age, parity, education, marital status, household income (minimum salary average guaranteed by law in Brazil is USD194 per month), habits (smoking and alcohol consumption), pregnancy complications (hypertensive disorders and diabetes), companionship during labor, gestational age, mode of delivery, birth weight, 5-min Apgar score, history of COVID-19 during pregnancy and history of COVID-19 in the family.

Next, the participants were asked to complete a questionnaire with statements addressing their knowledge and concerns about the pandemic, including information on prenatal care, and recommendations for childbirth and postpartum (including breastfeeding). This questionnaire comprised four domains: general knowledge and preventive care (4 items), prenatal concerns (4 items), cautions and fears during childbirth (5 items), care and concerns about the newborn (5 items). Each item received a 5-point Likert response ranging from 1 to 5 (strongly disagree, partially agree, indifferent, partially agree and fully agree). This questionnaire was specifically designed for the study, with input provided by the involved researchers.

The Beck Anxiety Inventory (BAI) was used to measure maternal anxiety. The questions were answered following brief and standard instructions. The BAI consists of a 21-self-reported-item questionnaire for assessing anxiety level. Each item describes a common symptom of anxiety and is rated on a 4-point Likert scale ranging from 0 (not at all) to 3 (severe). The respondent was asked to rate each symptom and then the total score was calculated (0–63). A high overall score indicates a high level of anxiety. Anxiety levels are defined according to the total score as follows: minimal anxiety (0–7), mild anxiety (8–15), moderate anxiety (16–25), and severe anxiety (26–63) [17]. A validated Brazilian Portuguese version of the BAI with adequate internal consistency (0.88–0.92) was used in this study [18,19].

### 2.3. Statistical Analysis

Data were analyzed using the MedCalc^®^ Statistical Software version 19.5.3 (MedCalc Software Ltd., Ostend, Belgium; 2020). Descriptive statistics are presented as mean and standard deviation (SD), median (95% CI), or frequency and percentage (%). The associations of categorical variables with binary outcomes were analyzed using Fisher’s exact test or the chi-square test when appropriate. The Mann-Whitney U tests were applied to continuous variables with non-parametric distribution. The analyses were adjusted for potential confounders such as maternal age, nulliparity, ethnicity, educational level, marital status (yes/no), religious belief (yes/no), smoking (yes/no), alcohol consumption (yes/no) and geographical location. The logistic regression model was used to investigate which variables were independently associated with increased anxiety. Statistical significance was set at *p* < 0.05.

## 3. Results

During the three-month period, 1683 eligible women were invited to participate in the study: only 21 refused, with 1662 (98.8%) women completing the questionnaires. Maternal sociodemographic characteristics, perinatal data and the overall results of the BAI score are shown in Table 1. Most women were non-white and living with a partner, and the majority reported educational level up to high school. BAI scores in late pregnancy indicate that 13.9% presented moderate and 9.6% severe anxiety. There were 3.2% of participants diagnosed with COVID-19 previously during pregnancy and 5.2% with a family member diagnosed with COVID-19.

Maternal anxiety differed according to geographic region (Table 2). The BAI total score was significantly higher in women interviewed in the Central West, South and Southeast than those in the North and in the Northeast. The Northeast exhibited the lowest scores of all the locations. The Central West region had the highest proportion of moderate or severe maternal anxiety (39% of respondents) and the Northeast the lowest proportion (8.9%).

The crude and adjusted analyses for confounding factors of the severity of maternal anxiety are shown in Table 3 and Table 4. Table 3 shows that the variables “cohabiting with a partner” (aOR 0.53, 95% CI 0.38–0.75) and “being from the North (aOR 0.49, 95% CI 0.32–0.77), Northeast (aOR 0.14, 95% CI 0.09–0.21) or Southeast” (aOR 0.57, 95% CI 0.40–0.82) were protective factors for maternal anxiety. The variables “secondary educational level” (aOR 1.66, 95% CI 1.21–2.29), “alcohol consumption” (aOR 3.5, 95% CI 1.94–6.14) and “having a family member diagnosed with COVID-19” (aOR 1.88, 95% CI 1.11–3.16) were independent factors, significantly associated with moderate or severe maternal anxiety at the end of pregnancy. Table 4 shows that protective factors for severe maternal anxiety were “cohabiting with a partner” (aOR 0.51, 95% CI 0.32–0.79) and “being from the Northeast” (aOR 0.17, 95% CI 0.09–0.31), and that “secondary education” (aOR 2.14, 95% CI 1.30–3.54) or “university educational level” (aOR 2.36, 95% CI 1.26–4.43) were independently associated with severe maternal anxiety.

Furthermore, women’s knowledge and concerns about the COVID-19 pandemic were related to moderate or severe maternal anxiety as assessed by the total BAI score (Table 5). Knowing the signs and symptoms of the disease and having confidence in self-protection and adequate guidance on breastfeeding were aspects associated with anxiety. Moderate or severe maternal anxiety was independently associated with the fear of being unaccompanied at childbirth (aOR 1.12, 95% CI 1.10–1.35). Independent protective factors were confidence in knowing how to protect oneself from COVID-19 (aOR 0.89, 95% CI 0.82–0.97) and how to breastfeed during the COVID-19 pandemic (aOR 0.89, 95% CI 0.83–0.95).

## 4. Discussion

To the best of our knowledge, this is the first study in Brazil to investigate maternal anxiety in late pregnancy during the COVID-19 outbreak and to examine its association with knowledge and concerns regarding childbirth. Our study found moderate or severe anxiety in 23.4% of the women at the end of pregnancy. Moderate or severe maternal anxiety was related to social factors such as educational level, living with a partner, alcohol consumption and having a family member diagnosed with COVID-19. Concerns that increased maternal anxiety were mainly related to confidence in self-protection from COVID-19, doubts regarding the presence of a companion during childbirth and safety of breastfeeding.

Preventive actions and treatment of COVID-19 in the obstetric scenario are daunting tasks requiring the optimization of both the mother and the baby’s health care. The potential effects of maternal infection on the fetus and the newborn are a source of concern and insecurity [20]. COVID-19 during pregnancy is associated with an increased risk of adverse maternal and perinatal outcomes and severe disease [21]. In addition, in Brazil, Takemoto et al. [22] reported increased maternal deaths and delays associated with health care: a large proportion of deaths occurred without intensive care or ventilatory support. They found a case fatality rate of 12.7% among women with severe acute respiratory syndrome due to COVID-19 during pregnancy and the postpartum period.

Anxiety status and the factors influencing it may differ according to the severity of the outbreak in each geographic region. In the present study, data were collected from all geographic regions in Brazil during the same time period. Maternal anxiety was more prevalent and more severe in the Central West and in the South. In the months the interviews were conducted, the number of COVID-19 cases was still low in the North and in the Northeast and was increasing in the South and in the Central West. This may have influenced the prevalence of mental health disorders in the women. The notifications of deaths could potentially impact maternal mental health, and the risk of anxiety disorders may have increased as a result. The same was reported by Liu et al. [14], who found differences between cities: more women in Wuhan (epicenter) felt anxious than in Chongqing, a less affected city (24.5% vs. 10.4%). Brazil is a country of continental size, with great disparities among regions, not only in the number of COVID-19 cases, but also in social, cultural and economic characteristics.

Anxiety due to COVID-19, as assessed by a specific instrument, significantly influenced social attitudes [23]. The prevalence of psychological problems associated with social isolation during the COVID-19 epidemic in China was verified through an online survey, which specifically detected high rates of anxiety, depression, alcohol consumption and poor mental well-being among Chinese people [24]. Pregnant women assessed after the declaration of the COVID-19 epidemic in China had mean depressive scores and anxiety subscale scores which were significantly higher than those before the declaration. The higher risk for depressive symptoms included people with an annual middle household income and low levels of physical exercise [15]. In the present study, the role of family income was not analyzed but, contrary to the finding of Wu et al. [15], higher educational levels increased significantly the risk of severe maternal anxiety.

The uncertainty about the best treatment and clinical management of patients with COVID-19 may affect the mind and psyche of pregnant women, with implications for their health care experiences [25]. Depression and anxiety are associated, and both can occur due to the COVID-19 pandemic [26]. Uncertain prognosis, social isolation, restriction of individual freedoms, financial losses, decline in quality of life and conflicting messages from government authorities may add up as stressors and possibly trigger mental health crises.

At the hospital, childbirth care must prioritize maternal safety. The removal of personal items, the reduction in the number of visitors and companions, the reduction in the team assisting childbirth, the use of masks and personal protective equipment by all professionals are measures that possibly impact the expectations [27] of these pregnant women about the experience of childbirth. In Italy, Ravaldi et al. [28] reported birth expectations completely changed after the onset of COVID-19 to include fear, sadness and uncertainty. Ahlers–Schmidt et al. [29] carried out an electronic survey with a 51% response rate of pregnant women or mothers of infants less than 12 months, 82.5% of whom reported changes in mental status, including an increase of anxious thoughts related to the COVID-19 pandemic of 50%. In Italy, almost half of the women (46%) in an interview reported high anxiety regarding risks of vertical transmission of the disease [30]. In the present study, we found that maternal anxiety was associated with concerns related to the moment of delivery and breastfeeding as a consequence of the pandemic.

An online survey investigated maternal mental health status after a few weeks of lockdown in Belgium using the Generalized Anxiety Disorder 7-item Scale [16]. It revealed higher levels of overall anxiety among pregnant women, 8.4% of whom had moderate anxiety and 5.2% of whom had severe anxiety. The social isolation measures brought on by COVID-19 may be a great burden on the emotional well-being of women in the initial postpartum period. We found 13.9% cases of moderate anxiety and 9.6% cases of severe anxiety. We also found that insecurity with respect to breastfeeding and fear of being unaccompanied at childbirth were associated with higher scores on the total BAI, indicating that counseling for such issues should be improved during antenatal care and at hospital admission.

Nodoushan et al. [31] evaluated the spiritual and mental health of pregnant women and COVID-19 anxiety during the pandemic in Iran, and found direct and significant association. In Pakistan, 36% of pregnant women interviewed stated the perception that the COVID-19 pandemic had a large impact on their mental health [32].

It is important for women to have adequate support, which includes health care workers and companions during labor and childbirth. It is up to midwives and obstetricians to help women regain self confidence and trust in caregivers. Educational programs should be designed to address pregnant women’s perceptions of the COVID-19 pandemic in order to improve their mental well-being.

In this study, the models were constructed with maternal anxiety categorized as minimal or mild versus moderate or severe and also not severe versus severe because this level of anxiety was considered relevant for the Brazilian population and the proposed model was intended to enable the evaluation of factors associated with increased anxiety. The relationship between severe anxiety and women’s educational level possibly indicates the influence of social inequalities on access to information and perception of severity.

### Strengths and Limitations

The strength of this study lies in the inclusion of 10 cities in all geographic regions in Brazil, most of which are state capitals, during the time when South America was the epicenter of COVID-19 pandemic in the world. Additionally, the women were interviewed face-to-face, not through online electronic forms or phone calls, and by trained doctors who were available to answer questions and minimize concerns. Our study has the limitation of including only participants from the public sector in university hospitals, which admit high- and low-risk pregnant women for childbirth. Patients who were suspected or confirmed cases of COVID-19 at admission for childbirth were not included. Had they been included, the results might have been different. In addition, when previous COVID-19 infection was reported during the interview, no detailed information was collected regarding the symptoms or diagnostic testing at the time of infection. Additionally, unfortunately, variables of postpartum depression or measure of optimism for any scale were not collected. Another limitation is that the emergence of COVID-19 was different in the diverse geographic regions. The disease started in the Southeast, the North and the Northeast. Furthermore, there were organizational differences in the health systems among cities, which may have influenced quality of health care during the period.

## 5. Conclusions

Many questions remain unanswered regarding the ongoing pandemic and a scenario of uncertainty continues. Highlighting mental health problems, especially anxiety among the general population and among pregnant women, is of paramount importance during public health emergencies. Our findings show that the impact of COVID-19 has been significant on maternal anxiety, especially with concerns about limitations for a companion during childbirth and breastfeeding safety. Lack of a partner, high educational level, consumption of alcohol and a family member with COVID-19 increased the risk for higher levels of anxiety. As Brazil is currently registering a new increase in COVID-19 cases, we urge authorities to prioritize women’s health and consider immediate interventions in order to provide recommendations to reduce the impacts of the pandemic on mental health.

## Figures and Tables

**Figure 1 jcm-10-00620-f001:**
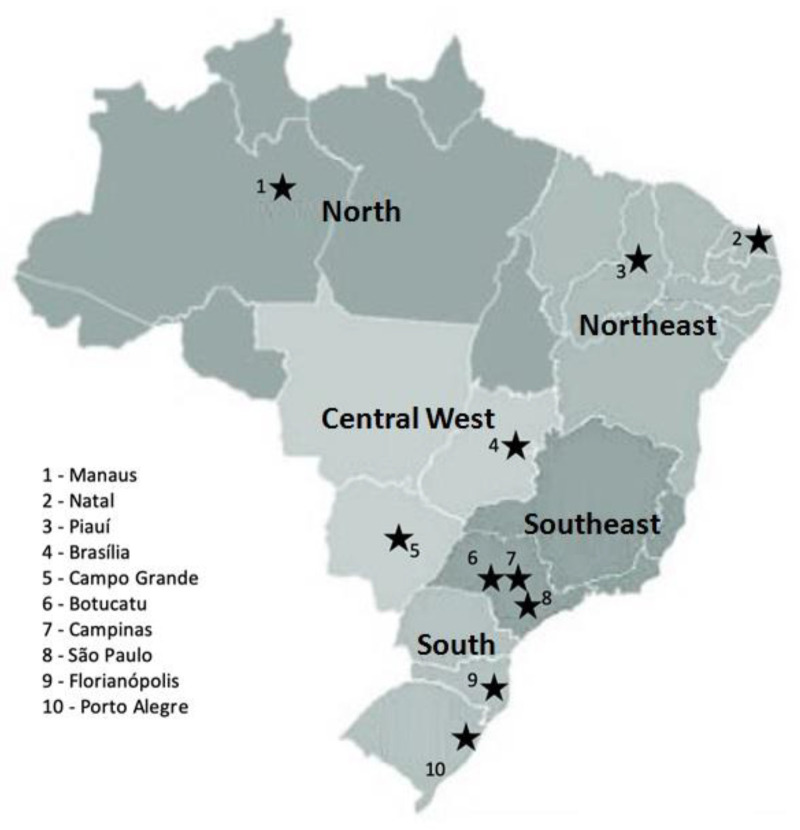
Map of Brazil with the location of the 10 cities in which the study was conducted.

**Table 1 jcm-10-00620-t001:** Characteristics, perinatal outcomes and maternal anxiety assessed by the Beck Anxiety Inventory (BAI) at the end of pregnancy during the Coronavirus disease (COVID-19) outbreak in Brazil (June–August, 2020). (*n* = 1662).

Characteristics	Results	
Maternal age, years, mean, SD	28.2	6.5
Parity 0	648	39.0%
Maternal ethnicity		
White	454	27.3%
Mixed	1020	61.4%
Black	166	10.0%
Asian or Brazilian Indian	22	1.3%
Cohabiting/married	1458	87.7%
Educational level		
Below primary school	12	0.7%
Primary school	423	25.5%
Secondary school	982	59.1%
College/university	245	14.7%
Lower class household income—less than USD194/month	1039	62.5%
Religious belief	1330	80.0%
Smoking	70	4.2%
Alcohol consumption	69	4.2%
Mode of delivery		
Vaginal	798	48.0%
Cesarean	843	50.7%
Forceps/vacuum	21	1.3%
Companionship in labor	1149	69.1%
Gestational age at birth, weeks, mean, SD	39.1	1.3
Birth weight, g, mean, SD	3298	502
Low birth weight (<2500 g)	81	4.9%
Macrosomia (>4000 g)	132	7.9%
5-min Apgar < 7	47	2.8%
COVID-19 during pregnancy	54	3.2%
COVID-19 in the family	86	5.2%
Maternal anxiety		
Minimal	900	54.2%
Mild	372	22.4%
Moderate	231	13.9%
Severe	159	9.6%
BAI, total score		
Mean, SD	9.9	10.4
Median (IQR)	6.0	12.0

**Table 2 jcm-10-00620-t002:** Maternal anxiety at end of pregnancy according to geographic regions during the COVID-19 outbreak in Brazil (June–August, 2020).

			Geographic Location									
	Total		Central West (*n* = 300)		North (*n* = 191)		Northeast (598)		South (*n* = 265)		Southeast (*n* = 308)	
Maternal anxiety (BAI)												
Minimal (0–7)	900	(54.2)	110	(36.7)	99	(51.8)	459	(76.8)	105	(39.6)	127	(41.2)
Mild (8–15)	372	(22.4)	73	(24.3)	48	(25.1)	86	(14.4)	69	(26.0)	96	(31.2)
Moderate (16–25)	231	(13.9)	70	(23.3)	26	(13.6)	34	(5.7)	52	(19.6)	49	(15.9)
Severe (26–63)	159	(9.6)	47	(15.7)	18	(9.4)	19	(3.2)	39	(14.7)	36	(11.7)
BAI total score												
Mean (SD)	9.9	(10.4)	13.9	(11.4)	10.2	(11.5)	5.3	(7.1)	13.0	(11.0)	11.8	(10.0)
Median (IQR) *	6.0	(12.0)	11	(15.0) ^a^	5	(11.0) ^b^	3	(7.0) ^c^	10	(14.0) ^d^	9	(13.0) ^e^

BAI: Beck Anxiety Inventory. * Kruskal–Wallis test *p* < 0.001, Post-hoc analysis (Conover): ^a^ CW different from N and NE, *p* < 0.05. ^b^ N different from CW, NE, S, SE, *p* < 0.05. ^c^ NE different from CW, N, S, SE, *p* < 0.05. ^d^ S different from N, NE, *p* < 0.05. ^e^ SE different from N, NE, *p* < 0.05.

**Table 3 jcm-10-00620-t003:** Characteristics of women at birth, complications during pregnancy, and geographic region in Brazil according to moderate or severe maternal anxiety as assessed by the Beck Anxiety Inventory during the COVID-19 outbreak (June–August, 2020).

Characteristic	Maternal Anxiety				
	Minimum or Mild (*n* = 1272)	Moderate or Severe (*n* = 390)	*p* Value ^a^	aOR (95% CI) ^b^	*p* Value ^b^
Age, years, median, IR	28 (23, 33)	27 (26, 28)	0.495	-	0.919
Parity 0	484 (38.1)	164 (42.1)	0.157	-	0.685
White ethnicity	310 (24.4)	144 (36.9)	<0.001	-	0.054
Black ethnicity	119 (9.4)	47 (12.1)	0.120	-	0.278
Cohabiting/married	1134 (89.2)	324 (83.1)	0.001	0.53 (0.38–0.75)	<0.001
Educational level					
Secondary school	715 (56.2)	253 (64.9)	0.002	1.66 (1.21–2.29)	0.002
College/university	178 (14.0)	67 (17.2)	0.121	-	0.129
No religious belief	250 (19.7)	82 (21.0)	0.554	-	0.366
Smoking	47 (3.7)	23 (5.9)	0.058	-	0.919
Alcohol consumption	42 (3.3)	27 (6.9)	0.002	3.5 (1.94–6.14)	<0.001
COVID-19 during pregnancy	38 (3.0)	16 (4.1)	0.277	-	0.625
COVID-19 in the family	55 (4.3)	31 (7.9)	0.005	1.88 (1.11–3.16)	0.019
Maternal complications					
Hypertensive disorders	435 (34.2)	92 (23.6)	<0.001	-	0.385
Diabetes mellitus ^c^	213 (16.7)	57 (14.6)	0.319	-	0.235
Geographic location					
Central West	183 (14.4)	117 (30.0)	-	-	-
North	147 (11.6)	44 (11.3)	<0.001	0.49 (0.32–0.77)	0.002
Northeast	545 (42.8)	53 (13.6)	<0.001	0.14 (0.09–0.21)	<0.001
South	174 (13.7)	91 (23.3)	0.252	-	0.144
Southeast	223 (17.5)	85 (21.8)	0.003	0.57 (0.40–0.82)	0.002

Data are presented as median (interquartile range) or number (percentage). ^a^ Chi-square test. ^b^ Logistic regression to identify independent variables. ^c^ Pregestational and gestational diabetes mellitus. Variables considered for adjusted OR: age, parity, maternal race, cohabitating, educational level, religious belief, smoking, alcohol consumption, COVID-19 during pregnancy, COVID-19 in the family, maternal complications and geographic location. aOR: adjusted odds ratio; CI: confidence interval.

**Table 4 jcm-10-00620-t004:** Characteristics of women, complications during pregnancy, and Brazilian geographic regions according to severe maternal anxiety as assessed by the Beck Anxiety Inventory during the COVID-19 outbreak (June–August, 2020).

Characteristics	Maternal Anxiety				
	Not Severe (*n* = 1503)	Severe (*n* = 159)	*p* Value ^a^	aOR (95% CI) ^b^	*p* Value ^b^
Age, years, median, IR	27 (23, 33)	27 (22, 32)	0.209	-	0.2588
Parity 0	579 (38.5)	69 (43.4)	0.231	-	0.9643
White ethnicity	392 (26.1)	62 (39.0)	<0.001	-	0.2193
Black ethnicity	146 (9.7)	20 (12.6)	0.252	-	0.3257
Cohabiting/married	1330 (88.5)	128 (80.5)	0.004	0.51 (0.32–0.79)	0.0030
Educational level					
Secondary school	863 (57.4)	105 (66.0)	<0.001	2.14 (1.30–3.54)	0.0030
College/university	213 (14.2)	32 (20.1)	<0.001	2.36 (1.26–4.43)	0.0074
No religious belief	299 (19.9)	33 (20.8)	0.796	-	0.5077
Smoking	59 (3.9)	11 (6.9)	0.074	-	0.3505
Alcohol consumption	61 (4.1)	8 (5.0)	0.559	-	0.3008
COVID-19 during pregnancy	46 (3.1)	8 (5.0)	0.183	-	0.9624
COVID-19 in the family	71 (4.7)	15 (9.4)	0.011	-	0.0745
Maternal complications					
Hypertensive disorders	485 (32.3)	42 (26.4)	0.132	-	0.0718
Diabetes mellitus ^c^	245 (16.3)	25 (15.7)	0.851	-	0.8603
Geographic location					
Central West	253 (16.8)	47 (29.6)	-	-	-
North	173 (11.5)	18 (11.3)	0.047	-	0.2317
Northeast	579 (38.5)	19 (11.9)	<0.001	0.17 (0.09–0.31)	<0.0001
South	226 (15.0)	39 (24.5)	0.754	-	0.6892
Southeast	272 (18.1)	36 (22.6)	0.154	-	0.1748

Data are presented as median (interquartile range) or number (percentage). ^a^ Chi-square test. ^b^ Logistic regression to identify independent variables. ^c^ Pregestational and gestational diabetes mellitus. Variables considered to be adjusted the OR: age, parity, maternal race, cohabitating, educational level, religious belief, smoking, alcohol consumption, COVID-19 during pregnancy, COVID-19 in the family, maternal complications and geographic location. aOR: adjusted odds ratio; CI: confidence interval.

**Table 5 jcm-10-00620-t005:** Women’s knowledge and concerns about the COVID-19 pandemic associated with moderate or severe maternal anxiety as assessed by the Beck Anxiety Inventory (June–August, 2020).

	Minimum or Mild (*n* = 1272)		Moderate or Severe (*n* = 390)				
	Median	Average Rank	Median	Average Rank	*p* Value ^a^	aOR (95% CI) ^b^	*p* Value ^b^
**Knowledge and preventive care**						**-**	**-**
I am afraid of getting COVID-19.	5	834.5	5	821.8	0.581	-	-
I know the signs and symptoms of COVID-19.	5	851.5	4	766.3	0.001	-	-
I received information about care in the pandemic.	5	826.9	5	846.3	0.350	-	-
I feel confident in protecting myself from COVID-19.	5	855.0	4	755.0	0.000	0.89 (0.82–0.97)	0.007
**Prenatal concerns**							
I was worried about COVID-19 affecting my baby during pregnancy.	5	824.5	5	854.2	0.118	-	-
I was instructed on caring for the newborn.	5	836.8	5	814.3	0.369	-	-
I was guided on breastfeeding during COVID-19.	5	828.4	4.5	841.7	0.600	-	-
I was worried about prenatal difficulties.	5	828.0	5	842.9	0.514	-	-
**Cautions and fears during childbirth**							
I received guidance on childbirth care due to COVID-19.	4	849.8	4	772.0	0.003	-	-
My companion was afraid of COVID-19 at delivery.	5	837.2	5	813.0	0.337	-	-
I was worried about giving birth at the hospital.	5	823.5	5	857.5	0.183	-	-
I was afraid to be without a companion in childbirth.	5	809.6	5	902.9	<0.0001	1.12 (1.10–1.35)	<0.001
I was worried that childbirth care might be compromised due to COVID-19.	5	824.1	5	855.5	0.218	-	-
**Care and concerns about the newborn**							
I learned how to breastfeed due to COVID-19.	5	858.0	4	745.1	<0.0001	0.89 (0.83–0.95)	0.001
I feel confident to breastfeed despite COVID-19.	5	854.6	5	756.3	<0.0001	-	-
I’m worried about having COVID-19.	5	836.3	5	815.8	0.382	-	-
I’m worried that my baby has COVID-19.	5	841.4	5	799.3	0.089	-	-
I’m worried about my baby having COVID-19 after birth.	5	825.7	5	850.5	0.261	-	-

^a^ Mann–Whitney test. ^b^ Logistic regression with stepwise procedure to select independent variables. aOR: adjusted odds ratio; CI: confidence interval.

## Data Availability

The data presented in this study are available on request from the corresponding author. The data are not publicly available due to privacy restrictions.

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
