# Peer review of "Impact of the COVID-19 Pandemic on Maternal Anxiety in Brazil"

_jcm, 2021, doi:10.3390/jcm10040620_

Round 1

Reviewer 1 Report

The Nomura et al article was a great and interesting work related to maternal anxiety in late pregnancy promoted by COVID-19 in a Brazilian population. The maternal anxiety is inherent in new mothers, which also is linked with other psychological variable as a depression, because the maternal context and priorities change per se. With world lockdown by COVID-19, the fears and concerns increasing in general population and, in pregnancy women could be a risk population to be affected. In addition, the anxiety during pregnancy could be associated with nursery care of the offspring, such as breastfeeding strategy or attachment child behavior. However, some kindly suggestions would do to improve the compressibility of the text.      

Mayor comments:

  • Introduction section:
    • Line 67 the authors could report the prevalence of depression and anxiety in these regions before pandemic to highlights the effect of the lockdown.
    • It would be useful if the line 70 would have a reference. In addition, the prevalence of mental health in Hong Kong cut the speech. Some sentence to introduce the importance of these data would be necessary. Furthermore, the paragraph from 76 to 86 summaries the same idea that before, I kindly suggest merge both paragraph in one.
    • Line 88, what authors mean with vertical transmission related to anxiety?
  • Material and Methods section:
    • Could the authors give some example for “psychiatric or mental disorders; and in good clinical condition”? in addition, the min. salary average could have some reference to justify this threshold in the Brazil context.
    • The variables of “Religious belief”, “Smoking” and “Alcohol consumption” need to be define. Was it binary (yes/no)? what religious? Smoking habits could be huge, non-smoking, passive-smoking, leave-2-years-smookers…? Same for the alcohol intake.
    • In BAI scale, previous internal consistence in Brazilian or pregnant need to be reported.
    • For the median, the IQR could be a better measurement of dispersion in no-normal distribution variable.
  • Result section:
    • The sex, birth weight z-score and SNAP-II was collected? If yes, need to be informed. In addition, when the COVID-19 positive test during pregnancy was done could be interesting, and also if the mother was symptomatic or asymptomatic because could not be the same pass the disease asymptomatic in the 10 week of gestation than symptomatic in the 32 week of gestation.  
    • Were general resources, postpartum depression or optimism measurement for any scale (i.e. LOT, EPDS)? These variables have been demonstrated affect the pregnancy mental health. That variables could be described in the discussion section as a limitation of the study.
    • Table 2. I kindly suggest if the authors could implement the significant inside the table (i.e. different letter to say significant differences). Also, the table could update with the prevalence of COVID-19 in these regions according Brazil reports.
    • For table 3 and 4, the footnote should write what the variables were considered to be adjusted the OR in the models.
  • Discussion section:
    • Lines 237-248. Great and useful paragraph.
    • The authors need to clarify, why did they built models with maternal anxiety categorized in Minimum or mild versus Moderate or severe and also not-severe versus severe, what more information report the table 4 compared to table 3?
  • Conclusion section:
    • The authors did not measurement depression. In addition, according to their results high educational level could be a protective factor among COVID-19, it could be confusing (also lines 257-258).

Minor comments:

  • Double check the journal guidelines about the text and styles.
  • Carefully suggest included subheading in the material and methods section (i.e. women enrollment, questionnaires application and statistical analysis)
  • Table 3, the difference between maternal race (white/black) and maternal anxiety category (moderate/severe) would be a 2x2 table with one p-value associated in the univariate analysis. Please double-check.
  • Clarify if the diabetes mellitus was gestational diabetes.
  • Line 249. “Anxiety due to the coronavirus” should be more specifically, because coronavirus is a family including other virus such as MERS.

Author Response

Dear editor,

We would like to thank you for the opportunity to revise the manuscript entitled “Impact of the COVID-19 pandemic on maternal anxiety in Brazil” submitted to JCM - Journal of Clinical Medicine. We would also like to thank the reviewers for the positive feedback and helpful comments that greatly helped to improve the manuscript. We look forward to working with you and the reviewers to move this manuscript toward publication in the JCM. We have detailed the changes or responses to Reviewer 1’s and Reviewer 2’s comments below. Our responses to the reviewer´s comments are provided after each comment.

#Reviewer 1

The Nomura et al article was a great and interesting work related to maternal anxiety in late pregnancy promoted by COVID-19 in a Brazilian population. The maternal anxiety is inherent in new mothers, which also is linked with other psychological variable as a depression, because the maternal context and priorities change per se. With world lockdown by COVID-19, the fears and concerns increasing in general population and, in pregnancy women could be a risk population to be affected. In addition, the anxiety during pregnancy could be associated with nursery care of the offspring, such as breastfeeding strategy or attachment child behavior. However, some kindly suggestions would do to improve the compressibility of the text.      

Response: We greatly appreciate your comments and suggestions. Please find below a point-by-point answer to the questions raised regarding our submission of the revised manuscript.

Mayor comments:

Introduction section:

  1. Line 67 the authors could report the prevalence of depression and anxiety in these regions before pandemic to highlights the effect of the lockdown.

Response: Thank you for your comments. We have included this information to enable comparison and to show the increase in anxiety during the pandemic. Line 72

“Among the pregnant women evaluated in the primary care in Sao Paulo, reported anxiety prevalence was 20.4% before COVID-19 pandemic.”

  1. It would be useful if the line 70 would have a reference. In addition, the prevalence of mental health in Hong Kong cut the speech. Some sentence to introduce the importance of these data would be necessary. Furthermore, the paragraph from 76 to 86 summaries the same idea that before, I kindly suggest merge both paragraph in one.

Response: Thank you for your comments. We agree and the reference was included. We agree that the text was truncated. We excluded the first 5 sentences of the paragraph and we have changed the text, to avoid repetition.

  1. Line 88, what authors mean with vertical transmission related to anxiety?

Response: Thank you for your comment, we agree it was a bit confusing and we decided to remove this sentence. The paragraph now reads:

“The effects of the pandemic in low- and middle-income settings are underreported. Mental health in pregnant women, during the period of greatest mortality in Brazil due to COVID-19 has not been assessed. Therefore, the aim of the present study was to evaluate the prevalence of maternal anxiety in late pregnancy in the context of the COVID-19 outbreak in Brazil and to investigate the association with sociodemographic characteristics and maternal knowledge and concerns about the pandemic.”

Material and Methods section:

  1. Could the authors give some example for “psychiatric or mental disorders; and in good clinical condition”? in addition, the min. salary average could have some reference to justify this threshold in the Brazil context.

Response: Thank you for your comments. We have included examples in line 118: “…absence of psychiatric or mental disorders (depression, bipolar disorder, schizophrenia, etc.); and in good health, in the absence of severe obstetrical/clinical complication during childbirth or Intensive Care Unit (ICU) admission …”. We have included the information about household income, as this minimum salary is guaranteed by law in Brazil.

  1. The variables of “Religious belief”, “Smoking” and “Alcohol consumption” need to be define. Was it binary (yes/no)? what religious? Smoking habits could be huge, non-smoking, passive-smoking, leave-2-years-smookers…? Same for the alcohol intake.

Response: Thank you for your comments. All these variables were investigated as binary (yes/no) and we included this information in the text. In Brazil, religious syncretism is huge, as we don't only have 2 or 3 religions, including some African religions mixed in our culture, so we decided to investigate this aspect as a binary variable of having or not having any religious belief. We acknowledge that more details into these variables could be interesting, however, it was not the scope of this analysis.

  1. In BAI scale, previous internal consistence in Brazilian or pregnant need to be reported.

Response: Thank you for your comments. The psychometric properties of the Brazilian Portuguese version of the BAI showed adequate internal consistency (0.88-0.92). We have included this information.

  1. For the median, the IQR could be a better measurement of dispersion in no-normal distribution variable.

Response: Thank you for your comments. We have changed the measurement to IQR.

Result section:

  1. The sex, birth weight z-score and SNAP-II was collected? If yes, need to be informed. In addition, when the COVID-19 positive test during pregnancy was done could be interesting, and also if the mother was symptomatic or asymptomatic because could not be the same pass the disease asymptomatic in the 10 week of gestation than symptomatic in the 32 week of gestation.  

Response: Thank you for your comments. Unfortunately these data were not collected. Also, unfortunately, no information was collected regarding the symptoms of pregnant women at the time of COVID-19. We understand that this is a limitation of the study, and this information has been included in the discussion section. Line 318: “In addition, unfortunately, when previousCOVID-19 infection was reported during the interview, no detailed information was collected regarding the symptoms or diagnostic testing at the time of infection.”

  1. Were general resources, postpartum depression or optimism measurement for any scale (i.e. LOT, EPDS)? These variables have been demonstrated affect the pregnancy mental health. That variables could be described in the discussion section as a limitation of the study.

Response: Thank you for your comments.   Unfortunately, data on depression and optimism were not collected. This limitation was included in the discussion section. Line 318: “Also, unfortunately, variables of postpartum depression or measure of optimism for any scale were not collected.”

  1. Table 2. I kindly suggest if the authors could implement the significant inside the table (i.e. different letter to say significant differences). Also, the table could update with the prevalence of COVID-19 in these regions according Brazil reports.

Response: Thank you for your comments. We have added letters as suggested. Unfortunately, we do not have reliable data regarding the prevalence in the different regions, at that time of data collection, and we were unable to add this information. The country has faced great limitations in testing and official the surveillance system updates numbers of confirmed cases and deaths (information reported in the background)

  1. For table 3 and 4, the footnote should write what the variables were considered to be adjusted the OR in the models.

Response: Thank you for your comments. We have added this information.

Discussion section:

  1. Lines 237-248. Great and useful paragraph.

Response: Thank you for your comments.

  1. The authors need to clarify, why did they built models with maternal anxiety categorized in Minimum or mild versus Moderate or severe and also not-severe versus severe, what more information report the table 4 compared to table 3?

Response: Thank you for your comments. We have included a comment about it. Lines 303-306: “In this study, the models were constructed with maternal anxiety categorized as minimal or mild versus moderate or severe and also not severe versus severe because this level of anxiety was considered relevant for the Brazilian population and the proposed model was intended to enable the evaluation of factors associated to increased anxiety. The relationship between severe anxiety and women’s educational level possibly indicates the influence of social inequalities on access to information and perception of severity.”

Conclusion section:

  1. The authors did not measurement depression. In addition, according to their results high educational level could be a protective factor among COVID-19, it could be confusing (also lines 257-258).

Response: Thank you for your comments. We agree and the depression was excluded for conclusions section. Our data showed higher educational level related to severe anxiety and this comment was included.

Minor comments:

  1. Double check the journal guidelines about the text and styles.

Response: Thank you for your comments. We have checked the guidelines and the template.

  1. Carefully suggest included subheading in the material and methods section (i.e. women enrollment, questionnaires application and statistical analysis): women enrollment, questionnaires application, statistical analysis

Response: Thank you for your comments. We have included subheadings as suggested.

  1. Table 3, the difference between maternal race (white/black) and maternal anxiety category (moderate/severe) would be a 2x2 table with one p-value associated in the univariate analysis. Please double-check.

Response: Thank you for your comments. In Brazil we have a great racial miscegenation and these two white / black categories are the extremes of social inequalities, so they were analyzed separately.

  1. Clarify if the diabetes mellitus was gestational diabetes.

Response: Thank you for your comments. We have included in the table.

  1. Line 249. “Anxiety due to the coronavirus” should be more specifically, because coronavirus is a family including other virus such as MERS.

Response: Thank you for your comments. We have corrected to “Anxiety due to the COVID-19”

Reviewer 2 Report

First of all I want to congratulate the authors of the article because it
is a very important topic currently. Studies on COVID-19 are being of great use to the scientific community.
The analysis of anxiety is going to be a recurring theme because it will be present, not only during the pandemic, but once it is finished we will find many cases throughout our planet.
That said, I only have one suggestion for the authors. In the text of the article they use "race" as a variable. I do not know if it is an adequate term in the Brazilian context, but I would propose the change to the term "ethnicity".
Finally, I encourage the work team to continue investigating this topic in order to establish intervention programmes that favour the non-appearance of maternal anxiety.

Author Response

Dear editor,

We would like to thank you for the opportunity to revise the manuscript entitled “Impact of the COVID-19 pandemic on maternal anxiety in Brazil” submitted to JCM - Journal of Clinical Medicine. We would also like to thank the reviewers for the positive feedback and helpful comments that greatly helped to improve the manuscript. We look forward to working with you and the reviewers to move this manuscript toward publication in the JCM. We have detailed the changes or responses to Reviewer 1’s and Reviewer 2’s comments below. Our responses to the reviewer´s comments are provided after each comment.

#Reviewer 2:

  1. First of all I want to congratulate the authors of the article because it is a very important topic currently. Studies on COVID-19 are being of great use to the scientific community. The analysis of anxiety is going to be a recurring theme because it will be present, not only during the pandemic, but once it is finished we will find many cases throughout our planet.

Response: We greatly appreciate your encouragement as well as your comments

  1. That said, I only have one suggestion for the authors. In the text of the article they use "race" as a variable. I do not know if it is an adequate term in the Brazilian context, but I would propose the change to the term "ethnicity".

Response: Thank you for your comments. We have corrected it.

  1. Finally, I encourage the work team to continue investigating this topic in order to establish intervention programmes that favour the non-appearance of maternal anxiety.

Response: Thank you for your comments.